

# Use of micro and macroalgae extracts for the control of vector mosquitoes

Ozge Tufan-Cetin[1] and Huseyin Cetin[2]

[1] Department of Environmental Protection Technology, Vocational School of Technical Sciences, Akdeniz University, Antalya, Türkiye
[2] Department of Biology, Faculty of Science, Akdeniz University, Antalya, Türkiye

## ABSTRACT

Mosquitoes are one of the most dangerous vectors of human diseases such as malaria, dengue, chikungunya, and Zika virus. Controlling these vectors is a challenging responsibility for public health authorities worldwide. In recent years, the use of products derived from living organisms has emerged as a promising approach for mosquito control. Among these living organisms, algae are of great interest due to their larvicidal properties. Some algal species provide nutritious food for larvae, while others produce allelochemicals that are toxic to mosquito larvae. In this article, we reviewed the existing literature on the larvicidal potential of extracts of micro- and macroalgae, transgenic microalgae, and nanoparticles of algae on mosquitoes and their underlying mechanisms. The results of many publications show that the toxic effects of micro- and macroalgae on mosquitoes vary according to the type of extraction, solvents, mosquito species, exposure time, larval stage, and algal components. A few studies suggest that the components of algae that have toxic effects on mosquitoes show through synergistic interaction between components, inhibition of feeding, damage to gut membrane cells, and inhibition of digestive and detoxification enzymes. In conclusion, algae extracts, transgenic microalgae, and nanoparticles of algae have shown significant larvicidal activity against mosquitoes, making them potential candidates for the development of new mosquito control products.

Corresponding authors
Ozge Tufan-Cetin,
ozgetufan@akdeniz.edu.tr
Huseyin Cetin, hcetin@akdeniz.edu.tr

## INTRODUCTION

Mosquitoes are small flying insects that belong to the Culicidae family, which includes over 3,500 species. There are many genera of mosquitoes in the world, but the most important in terms of their impact on human health are the *Anopheles*, *Aedes*, and *Culex* genera. These insects are found all over the world, except for Antarctica, and are known for their annoying bites, which can cause itching and swelling (*Becker et al., 2003*; *Silver, 2008*). They are one of the most dangerous insects in the world, causing the transmission of deadly diseases such as malaria, dengue fever, chikungunya, Zika virus, yellow fever, and West Nile virus (*Suman et al., 2018*; *Moutinho et al., 2022*). Malaria is perhaps the most well-known mosquito-borne disease. It is caused by *Plasmodium* parasites that are transmitted through the bite of infected female *Anopheles* mosquitoes. This disease affects more than 200 million people each year and causes more than 400,000 deaths annually, majority of them being

children under the age of five (*WHO, 2022*). Mosquitoes need stagnant water to lay their eggs and for the larvae to develop. Therefore, any place that has stagnant water such as tires, septic tanks, ponds, lakes, and swamps can become a breeding site for mosquitoes (*Muhammad-Aidil et al., 2015*; *Alarcón-Elbal et al., 2021*).

Adult mosquitoes can be killed using insecticides that are applied by ground-based sprayers or aerial spraying in the form of spray, fog, or mist. Larvicides are insecticides that are used to control mosquito larvae. These insecticides can be applied to standing water or other breeding sites to prevent the development of adult mosquitoes. Both biological and chemical control methods can be used to manage mosquito populations, depending on the specific circumstances and goals of the mosquito control program (*Kumar & Sahgal, 2022*; *Medlock & Vaux, 2015*).

Biological control involves using natural enemies of mosquitoes to reduce their populations. For example, certain species of fish, such as *Gambusia affinis* and *Gam. holbrooki*, are known to feed on mosquito larvae and can be introduced into bodies of water to control mosquito populations. Bacterial insecticides, such as *Bacillus thuringiensis israelensis* (Bti), *Lysinibacillus sphaericus* and spinosad can also be used to target mosquito larvae (*Cetin, Yanikoglu & Cilek, 2005*; *Cetin, Dechant & Yanikoglu, 2007*). Chemical control involves using pesticides to kill adult mosquitoes or their larvae. For example, insecticides such as synthetic pyrethroids, carbamates and organophosphates can be sprayed in areas where mosquitoes are present to kill adult mosquitoes. Larvicides, such as pyriproxyfen and methoprene, can be applied to standing water to kill mosquito larvae (*WHO, 2013*).

Traditional mosquito control methods involve the use of chemical insecticides, but these approaches can have negative impacts on the environment and public health. Researchers have shown that chemical larvicides and adulticides such as methoprene, pyriproxyfen, imidacloprid and fipronil can negatively impact the growth and development of aquatic insects such as beetles, dragonflies, backswimmers and damselflies which are effective predators of mosquito larvae (*Moura & Souza-Santos, 2020*; *Lawler, 2017*; *Nakanishi et al., 2020*) and also, chemical insecticide resistance poses a significant challenge to mosquito control efforts, as it can reduce the effectiveness of insecticides and increase the costs associated with mosquito management. Resistance to larvicides and adulticides is a growing problem in the control of mosquito populations in the world (*Su & Cheng, 2012*; *Liu & Gourley, 2013*; *Ser & Cetin, 2015*; *Ser & Cetin, 2019*). Many vector mosquito species have developed resistance to insecticides through a variety of mechanisms including metabolic resistance, target-site resistance, behavioral resistance, penetration resistance and reduced sensitivity. To combat resistance, it is important to use a variety of control methods, including the rotation of different types of insecticides and the use of non-chemical methods such as mosquito nets and environmental management practices. Additionally, surveillance and monitoring programs should be implemented to detect and track resistance in mosquito populations (*Liu, 2015*; *Fotakis et al., 2022*).

For all these reasons, there is a need to accelerate efforts to explore alternative, environmentally friendly strategies to control mosquito populations. Alternative methods, such as the identification of biological control agents that utilize natural enemies of

mosquitoes, should be considered as a more sustainable and environmentally friendly solution (*Hamed, El-Sherbini & Abdeltawab, 2022*; *Rodrigues et al., 2022*). Algae are primary producers in aquatic ecosystems. Therefore, they can grow rapidly and produce various bioactive compounds. Since algae live in mosquito breeding areas, they are consumed as food by mosquitoes. Therefore, female mosquito species, such as *Culex quinquefasciatus*, choose areas containing microalgae as oviposition substrates (*Gil et al., 2021*). In addition, algae make important contributions to mosquito control by supporting mosquito predators such as dragonflies and aquatic insects in the habitat they create (*Chaves & Koenraadt, 2010*).

Microalgae and macroalgae are widespread throughout the world in a variety of species. Microalgae are small, single-celled organisms that can be found in a wide variety of environments such as oceans, lakes, rivers, and even soil. They are highly diverse and are found in both freshwater and marine ecosystems. Microalgae are known for their rapid growth and ability to adapt to different environmental conditions (*Borowitzka, 2013*). Macroalgae are large, multicellular algae found predominantly in marine environments. They grow attached to rocks, corals, or other substrates in intertidal and subtidal zones. Both microalgae and macroalgae are used as sources of food and nutrition. They are rich in essential nutrients, vitamins, minerals, and dietary fibers. Microalgae such as *Chlorella* sp. are used as dietary supplements and are known for their high protein content and beneficial fatty acids (*Spolaore et al., 2006*; *Laamanen et al., 2021*). Microalgae are also recognized as promising sources for the production of biofuels such as biodiesel, bioethanol and biogas (*Chisti, 2007*; *Mata, Martins & Caetano, 2010*). Brown macroalgae in particular are being studied as a potential source for biofuel production. Both microalgae and macroalgae are used in the production of various bioproducts, including pharmaceuticals, nutraceuticals, cosmetics, and bioplastics (*Plaza et al., 2010*; *Simonič & Zemljič, 2021*). They are extensively studied for their unique bioactive compounds, enzymes, and biomolecules. They have potential applications in medicine, biotechnology, and pharmaceutical research, including the development of new drugs, antimicrobial agents, and antioxidant compounds (*Jiménez-Escrig, Gómez-Ordóñez & Rupérez, 2011*; *Michalak & Chojnacka, 2015*).

Some algae species contain fatty acids and polyunsaturated fatty acids that are especially toxic to mosquito larvae. In this respect, we can refer to algae as natural producers of compounds with larvicidal activity against mosquito larvae. Research has also shown that some algae species produce compounds that inhibit the development of mosquito larvae. Therefore, using algae to control mosquito populations is a promising approach (*Menaa et al., 2021*; *Gil et al., 2021*). In this article, we reviewed the larvicidal potential of various algal extracts, transgenic microalgae, and nanoparticles of algae on mosquitoes and their underlying mechanisms (Fig. 1). We believe that this review will be of interest to professionals working in the fields of integrated pest management, biological control of mosquitoes, environmentally friendly bioinsecticides, algal secondary metabolites and the development of new algal-based biocides.

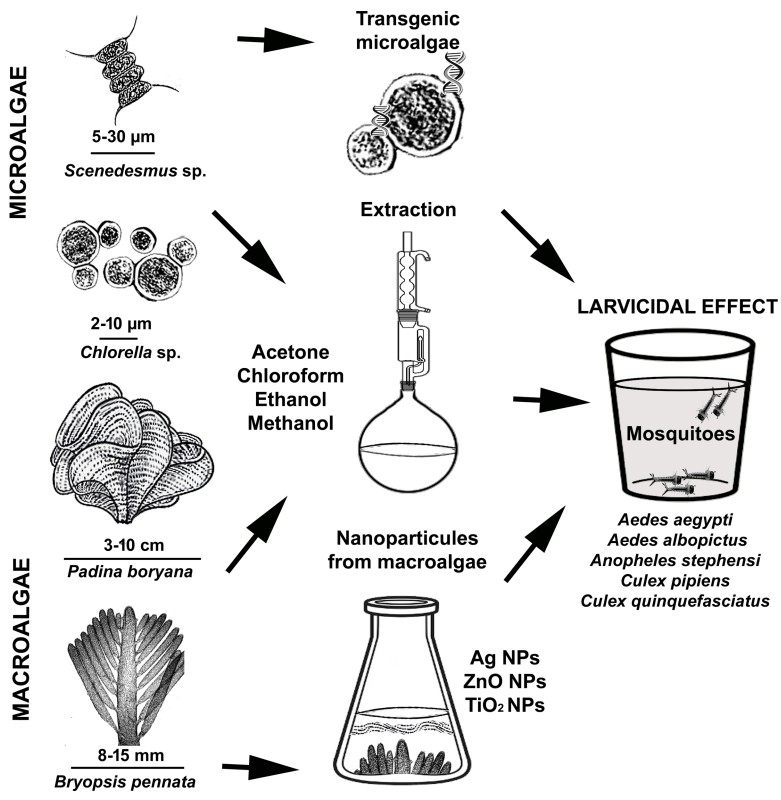

**Figure 1** Demonstration of using micro and macroalgae as mosquito larvicides.

## SURVEY METHODOLOGY

In this study, we conducted an up-to-date literature search covering the years 2010-2023 (until July 2023) on "Potential of micro and macroalgae extracts for larvicidal effect on mosquitoes" using various databases including PubMed, Scopus, Web of Science and Google Scholar. A limited number of pre-2010 studies have also been used in the evaluations, which contain information that supports the up-to-date study results. The keywords used for the search were "freshwater algae", "microalgae", "macroalgae", "transgenic algae", "mosquito control", "seaweed", "nanoparticles" and "larvicides". The criteria for inclusion in our article were that the articles discussed the use of micro and macroalgae for mosquito control. Although cyanobacteria are known as algae, they are prokaryotic photosynthetic bacteria. Therefore, this group of microorganisms was not included in this research. Articles with calculated lethal concentration values ($LC_{50}$ and/or $LC_{90}$) were primarily considered in our approach and evaluations, and these studies are listed in Tables 1 and 2. Studies using the Probit method as described by *Finney (1947)* were selected.

**Table 1  Larvicidal activity of microalgae extracts against mosquito larvae.**

| Microalgae | Mosquitoes /Larval instar | Extracts | LC$_{50}$ and LC$_{90}$ values at times | References |
|---|---|---|---|---|
| *Chlorella* sp. | *Aedes aegypti* /first instar | Chloroform | 116.8 and 312.9 ppm at 24 h | *Sigamani et al. (2020)* |
| *Chlorella* sp. | *Aedes aegypti* /second instar | Chloroform | 126.0 and 346.2 ppm at 24 h | *Sigamani et al. (2020)* |
| *Chlorella* sp. | *Aedes aegypti* /third instar | Chloroform | 132.7 and 547.1 ppm at 24 h | *Sigamani et al. (2020)* |
| *Chlorella* sp. | *Aedes aegypti* /fourth instar | Chloroform | 159.2 and 908.4 ppm at 24 h | *Sigamani et al. (2020)* |
| *Chlorella* sp. | *Aedes aegypti* /first instar | Methanol | 445.1 and 3,875.7 ppm at 24 h | *Sigamani et al. (2020)* |
| *Chlorella* sp. | *Aedes aegypti* /second instar | Methanol | 478.7 and 5,411.6 ppm at 24 h | *Sigamani et al. (2020)* |
| *Chlorella* sp. | *Aedes aegypti* /third instar | Methanol | 523.0 and 5,683.1 ppm at 24 h | *Sigamani et al. (2020)* |
| *Chlorella* sp. | *Aedes aegypti* /fourth instar | Methanol | 703.4 and 7,154.4 ppm at 24 h | *Sigamani et al. (2020)* |
| *Amphora coffeaeformis* | *Culex pipiens* /third instar | Acetone | 513.6 and 755.6 ppm at 24 h | *Hassan et al. (2021)* |
| *Scenedesmus obliquus* | *Culex pipiens* /third instar | Acetone | 855.6 and 1,277.4 ppm at 24 h | *Hassan et al. (2021)* |
| *Scenedesmus* sp. | *Aedes aegypti* /third instar | Ethanol | 514.2 and 1,053.0 ppm at 24 h | *Rani & Kumar (2023)* |
| *Scenedesmus* sp. | *Aedes aegypti* /third instar | Acetone | 746.4 and 1,380.2 ppm at 24 h | *Rani & Kumar (2023)* |
| *Scenedesmus* sp. | *Aedes aegypti* /third instar | Methanol | 735.3 and 1,320.1 ppm at 24 h | *Rani & Kumar (2023)* |
| *Chlorococcum* sp. | *Aedes aegypti* /third instar | Ethanol | 893.2 and 1,548.3 ppm at 24 h | *Rani & Kumar (2023)* |
| *Chlorococcum* sp. | *Aedes aegypti* /third instar | Acetone | 815.3 and 1,393.9 ppm at 24 h | *Rani & Kumar (2023)* |
| *Chlorococcum* sp. | *Aedes aegypti* /third instar | Methanol | 1,058.4 and 1,631.3 ppm at 24 h | *Rani & Kumar (2023)* |

# RESULTS AND DISCUSSION

## Mosquito larvicidal activity of microalgae extracts

Microalgae are unicellular photosynthetic microscopic algae, typically found in freshwater and saline environments. Microalgae have been searched for their potential use in various fields including biofuels, bioremediation, and as a source of valuable compounds such as pigments, proteins, and omega-3 fatty acids (*Nagi et al., 2021*; *Parmar et al., 2023*; *Xu, Miao & Wu, 2006*). When extracting compounds from algae, the choice of solvent and extraction method is crucial to ensuring efficient and effective extraction. The choice depends on the type of compounds targeted and their solubility properties. Solvents used in the extraction of algae are methanol, chloroform, ethanol, acetone, hexane, and water. The dried algae biomass is mixed or wetted with the selected solvent so that the solvent comes into contact

**Table 2  Larvicidal activity of macroalgae extracts against mosquito larvae.**

| Macroalgae | Mosquitoes /Larval instar | Extracts | LC$_{50}$ and/or LC$_{90}$ values at times | References |
|---|---|---|---|---|
| *Caulerpa scalpelliformis* | *Culex pipiens*/late second to early third instars | Acetone | 338.9 and 1,891.3 ppm at 72 h | *Cetin, Gokoglu & Oz (2010)* |
| *Dictyota linearis* | *Aedes aegypti* /fourth instar | Ethanol | 60 and 91.6 ppm at 24 h | *Bantoto & Dy (2013)* |
| *Padina minor* | *Aedes aegypti* /fourth instar | Ethanol | 50.8 and 84 ppm at 24 h | *Bantoto & Dy (2013)* |
| *Bryopsis pennata* | *Aedes aegypti* /third instar | Chloroform | 92.7 ppm at 24 h | *Yu et al. (2015)* |
| *Bryopsis pennata* | *Aedes aegypti* /third instar | Methanol | 156.9 ppm at 24 h | *Yu et al. (2015)* |
| *Bryopsis pennata* | *Aedes aegypti* /third instar | n-Hexane | 912.8 ppm at 24 h | *Yu et al. (2015)* |
| *Bryopsis pennata* | *Aedes albopictus* /third instar | Chloroform | 99.8 ppm at 24 h | *Yu et al. (2015)* |
| *Bryopsis pennata* | *Aedes albopictus* /third instar | Methanol | 177.5 ppm at 24 h | *Yu et al. (2015)* |
| *Bryopsis pennata* | *Aedes albopictus* /third instar | n-Hexane | 1,209.5 ppm at 24 h | *Yu et al. (2015)* |
| *Dictyota dichotoma* | *Aedes albopictus* /third instar | Ethanol | 44.32 and 85.92 ppm at 96 h | *Minicante et al. (2017)* |
| *Sargassum wightii* | *Anopheles stephensi* /third instar | Ethanol | 40.75 and 81.24 ppm at 24 h | *Suganya et al. (2019)* |
| *Sargassum wightii* | *Aedes aegypti* /third instar | Ethanol | 43.96 and 86.99 ppm at 24 h | *Suganya et al. (2019)* |
| *Sargassum wightii* | *Culex tritaeniorhynchus* /third instar | Ethanol | 47.83 and 90.96 ppm at 24 h | *Suganya et al. (2019)* |
| *Halimeda gracillis* | *Anopheles stephensi* /third instar | Ethanol | 52.95 and 106.58 ppm at 24 h | *Suganya et al. (2019)* |
| *Halimeda gracillis* | *Aedes aegypti* /third instar | Ethanol | 58.34 and 114.91 ppm at 24 h | *Suganya et al. (2019)* |
| *Halimeda gracillis* | *Culex tritaeniorhynchus* /third instar | Ethanol | 63.81 and 119.13 ppm at 24 h | *Suganya et al. (2019)* |
| *Champia parvula* | *Aedes aegypti* /third instar | Ethanol | 43 and 88 ppm at 48 h | *Yogarajalakshmi et al. (2020)* |
| *Porteria hornemannii* | *Culex quinquefasciatus* /fourth instar | Methanol | 2.4 ppm and 13.2 ppm at 48 h | *Arumugam et al. (2022)* |
| *Porteria hornemannii* | *Aedes aegypti* /fourth instar | Methanol | 33.1 ppm and 310.4 ppm at 48 h | *Arumugam et al. (2022)* |
| *Porteria hornemannii* | *Anopheles stephensi* /fourth instar | Methanol | 9.8 ppm and 117.0 ppm at 48 h | *Arumugam et al. (2022)* |
| *Porteria hornemannii* | *Culex quinquefasciatus* /fourth instar | Chloroform | 21.2 ppm and 335.9 ppm at 48 h | *Arumugam et al. (2022)* |
| *Porteria hornemannii* | *Aedes aegypti* /fourth instar | Chloroform | 116.1 ppm and 1,404.2 ppm at 48 h | *Arumugam et al. (2022)* |

**Table 2** (*continued*)

| Macroalgae | Mosquitoes /Larval instar | Extracts | LC$_{50}$ and/or LC$_{90}$ values at times | References |
|---|---|---|---|---|
| *Porteria hornemannii* | *Anopheles stephensi* /fourth instar | Chloroform | 68.5 ppm and 1,215.9 ppm at 48 h | *Arumugam et al. (2022)* |
| *Sargassum dentifolium* | *Culex pipiens* /third instar | Methanol | 284.3 and 1,032.1 ppm at 72 h | *Haleem et al. (2022)* |
| *Dictyota dichotoma* | *Culex pipiens* /third instar | Methanol | 227.3 and 857.9 ppm at 72 h | *Haleem et al. (2022)* |
| *Padina boryana* | *Culex pipiens* /third instar | Methanol | 253.8 and 991.6 ppm at 72 h | *Haleem et al. (2022)* |
| *Gelidium latifolium* | *Culex pipiens* /third instar | Methanol | 297.3 and 1,023.7 ppm at 72 h | *Haleem et al. (2022)* |
| *Jania rubens* | *Culex pipiens* /third instar | Methanol | 84.8 and 689.3 ppm at 72 h | *Haleem et al. (2022)* |
| *Galaxaura elongata* | *Culex pipiens* /third instar | Methanol | 31.1 and 400.8 ppm at 72 h | *Haleem et al. (2022)* |
| *Ulva intestinales* | *Culex pipiens* /third instar | Methanol | 97.5 and 755.6 ppm at 72 h | *Haleem et al. (2022)* |
| *Codium tomentosum* | *Culex pipiens* /third instar | Methanol | 69.8 and 556.5 ppm at 72 h | *Haleem et al. (2022)* |

with the algae cells and the desired compounds are extracted. Various extraction techniques such as maceration, infusion, decoction, and boiling under reflux, a wide range of modern techniques, such as microwave-extraction, ultrasound-extraction or supercritical $CO_2$ extraction can be used to improve extraction efficiency (*Stranska-Zachariasova et al., 2016*). More advanced techniques such as column chromatography, high-performance liquid chromatography, and solid-phase extraction can be applied to isolate or purify specific compounds.

Some metabolites synthesized by microalgae can be potential biological insecticides for the control of agricultural pests (*Costa et al., 2019*). Furthermore, research have shown that some microalgae species have larvicidal activity against mosquitoes and act on target organisms with the toxins they produce. Microalgae can also produce certain substances such as microcystin produced by cyanobacteria that inhibit larval development and delay the development of surviving larvae to the adult stage (*Rashed & El-Ayouty, 1991*; *Rey, Hargraves & O'Connell, 2009*). The utilization of microalgae for the control of mosquito larvae has gained significant attention due to their various advantages such as ease of production, cost-effectiveness, biodegradability, and being non-toxic to non-target organisms (*Asimakis et al., 2022*).

In one of the first remarkable studies on the subject before up-to-date studies (between 2010–2023), the effect of four green algae species, *Ankistrodesmus convolutus*, *Chlorella vulgaris*, *Chlorococcum* sp. and *Scenedesmus quadricauda* on some vital parameters of *Ae. aegypti* was investigated. It was found that mosquito larvae fed on *C. vulgaris, Chlorococcum* sp. and *S. quadricauda* were delayed in pupation and adults were reduced in size. In addition, at the end of the sixth exposure day, mortality range from 84–100% was detected in larvae fed with microalgae (*Ahmad et al., 2001*). After that research, the effect of ten green algal

isolates from mosquito breeding sites such as metal containers, discarded tires, and empty coconut shells on *Ae. aegypti* larvae for seven days was evaluated using a feeding test. Of these isolates, all mosquito larvae treated with *Chlorella* were found to die after seven days. The researchers interpreted the deaths of mosquito larvae as evidence for the resistance of these isolates to digestion. The authors also noted that the shape, size, and cell wall characteristics of algae may have an impact on their digestibility and consumption as food by mosquitoes (*Ahmad et al., 2004*). In a recent study, *Ae. aegypti* mosquito larvae fed *Chlorella sp.* were found to be affected in terms of development time, body size and life span. Larvae fed with the microalgae took longer to develop to pupation and adult stages, but their survival as adults was shorter (*Souza et al., 2019*).

Up-to-date articles in which microalgae species have been studied and proven to exhibit larvicidal activity against some mosquito species (indicating the solvent used and $LC_{50-90}$ values) are listed in Table 1. When the $LC_{50}$ and $LC_{90}$ values of mosquito species treated with microalgae extracts prepared with various solvent types in Table 1 are examined, the high lethal power of chloroform extracts is notable. It was reported in this research that when the chloroform extract of microalgae (*Chlorella* sp.) was purified, the larvicidal effect increased six times (*Sigamani et al., 2020*). It is seen from Table 1 that the larvicidal effect of chloroform has not been tested for other microalgae species (*Amphora coffeaeformis, Chlorococcum* sp., *Scenedesmus* sp., *Scenedesmus obliquus*) on mosquitoes. In a study using other solvents, the insecticidal activity of acetone, ethanol, and methanol extracts of *Chlorococcum* sp. and *Scenedesmus* sp. microalgae against third instar larvae of *Ae. aegypti* was investigated by *Rani & Kumar (2023)*. The authors reported that extracts of *Chlorococcum* sp. and *Scenedesmus* sp. showed larvicidal effect and the ethanol extract of *Scenedesmus* sp. had the highest larvicidal activity (Table 1).

In a comprehensive study with lethal concentration values indicated, the larvicidal potency of various solvent extracts of a microalgae species (*Chlorella* sp.) against four larval stages of the Dengue vector *Ae. aegypti* was examined by *Sigamani et al. (2020)*. Notably, purified fractions of chloroform extracts were found to have strong larvicidal activity against the mosquito larvae tested. The larvicidal activity in the fractions of chloroform extracts was attributed to the presence of n-hexadecanoic acid (palmitic acid), oleic acid, and β-sitosterol acetate compounds. Also in other studies, it has been reported that these substances have larvicidal properties (*Ragavendran, Dubey & Natarajan, 2017*; *Rahuman, Venkatesan & Gopalakrishnan, 2008*; *Rahuman et al., 2008*). *Sigamani et al. (2020)* emphasized that the high larvicidal activity was due to the particularly high n-hexadecanoic acid content in *Chlorella* sp. extract. In addition, they reported that third instar larvae of *Ae. aegypti* showed morphological and behavioral changes when exposed to a purified fraction of 50 ppm concentration of microalgae. In the study conducted using 200–1,000 ppm concentrations of extracts, it was found that younger stages of larvae were more susceptible than older ones.

In another research, *Hassan et al. (2021)* studied the potential insecticidal activity of two microalgae species: *Amphora coffeaeformis* and *Scenedesmus obliquus* against the larvae of *Cx. pipiens* mosquitoes. They reported that both species showed larvicidal activity, but *A. coffeaeformis* extract had a higher activity than *S. obliquus* extract. In the study, the presence

 

of substances such as polygalic acid, taxifolin, cinnamic acid, kaempferol, caffeic acid, methyl gallate, naringenin and syringic acid in the polyphenolic compound contents of these two species was pointed out. It was mentioned that the insect control effects of Gallic acid (toxic effect on many insects), Taxifolin (feeding deterrent), Cinnamic acid (high feeding inhibitor), Kaempferol (insecticidal effect), Caffeic acid (insect growth inhibitor and feeding deterrent), Naringenin (insecticidal effect) have been proven by previous studies.

The toxicity of aqueous extract and lectin purified from *Chlorella vulgaris* microalgae on *Ae. aegypti* fourth instar larvae was examined by *Cavalcanti et al. (2021)*. They showed that lectin from *C. vulgaris* inhibited the activity of trypsin-like enzymes in the larvae's gut. The $LC_{50}$ values of *C. vulgaris* aqueous extract and purified lectin after 72 h were 10.62% and 106.5 $\mu g\ ml^{-1}$, respectively. The authors stated that lectins are carbohydrate-binding proteins with previously proven insecticidal activity (*Oliveira et al., 2020*) and suggested that the purified lectin and the aqueous extract of *C. vulgaris* are potential mosquito larvicides. We hope that further research on this topic can be conducted and larvicide formulations can be developed.

*Rani & Kumar (2023)* reported that quinones and terpenoids detected in ethanol extracts of *Scenedesmus* sp. and *Chlorococcum* sp. but not found in other extracts may cause larvicidal effects. Similarly, saponins detected only in ethanol and methanol extracts may be the source of larvicidal activity. The insecticidal effect of quinones on mosquitoes has been proven in the past (*Silva et al., 2020*). In addition, terpenoid compounds derived from a species of red algae, dibromomertensene and dihydromertensene, have proven insecticidal effects (*Argandoña et al., 2000*). The mosquito-killing effect of these substances (quinones, terpenoids, saponins, lectins *etc.*) and their derivatives obtained from algae has not yet been studied and is thought to be the subject of many future studies.

Possible mechanisms of the larvicidal effect of algae on mosquitoes were reported to be based on damage to the gut of mosquito larvae by compounds in algae, inhibition of larval feeding, inhibition of digestive enzymes and processes, inhibition of ATPase production, inhibition of detoxification enzymes such as glutathione-s-transferase and synergistic effects of fatty acids and polyphenolic compounds (*Hassan et al., 2021*). Similar to this possible explanation of the mechanism of larvicidal activity, *Sigamani et al. (2020)* reported various histological changes in intestinal epithelial cells of larvae exposed to purified microalgal (*Chlorella* sp.) fractions of chloroform extracts compared to control groups. It was considered that in larvae exposed to microalgae fraction, compounds that enter the cell due to the damage of the exoskeleton bind to macro-components and cause physiological changes (*Farnesi et al., 2012*; *Dawet, Ikani & Yakubu, 2016*). Additionally, in one study, the levels of carboxylesterase (α and β), Glutathione-S-Transferase and Cytochrome P450, the major detoxifying enzymes of *Ae. aegypti* treated with algal extracts, were examined (*Yogarajalakshmi et al., 2020*). It was determined that there was a significant decrease in α and β carboxylesterase enzyme levels, whereas Glutathione-S-Transferase and Cytochrome P450 levels were higher in a dose-dependent manner. Thus, algal preparations administered to mosquitoes were found to cause significant physiological alterations by inhibiting or stimulating digestive enzyme activities.

## Mosquito larvicidal activity of macroalgae (or seaweed) extracts

Macroalgae, also known as seaweed, are chlorophyll-containing organisms. They are the source of some products such as alginate, carrageenan, and agar, all of which are deemed important in food and cosmetic industries (*Morais et al., 2021*). Polysaccharides, proteins, fatty acids, flavonoids, carotenoids and polyphenols in macroalgae have antifungal, antiviral, and antibacterial effects against some plant pathogens (*Hamed et al., 2018*). Many researchers have conducted studies showing that macroalgae have toxic effects on different insects, especially mosquitoes. *Yu et al. (2014)* compiled studies between 1991 and 2014 in which the effects of macroalgae extracts on some mosquito species were examined. Recent articles investigating the larvicidal activity of macroalgae extracts against some mosquito species are presented in Table 2.

The research in Table 1, *Bantoto & Dy (2013)* showed that the crude extracts of brown algae *Padina minor* and *Dictyota linearis* have larvicidal activity against fourth instar *Ae. aegypti* larvae. According to $LC_{50}$ values, *P. minor* showed significantly more larvicidal activity compared to *D. linearis*. It was reported that concentration-dependent mortality for both extracts and the mortality of *Ae. aegypti* larvae increased as the concentration of the extract increased. The researchers found that larvae exposed to the extracts had a longer larval stage than the control groups. It was also reported that the significant larvicidal activity of *Padina minor* and *Dictyota linearis* species against *Ae. aegypti* larvae may be due to their content of terpenoids, brown algal phenolics, unsaturated fatty acids or phlorotannins. Phlorotannins are a substance specific to brown algae and are rarely found in red algae. There is information in the literature that phlorotannins have a larvicidal effect, including on mosquitoes (*Negara et al., 2021*). Purified phlorotannins showed pesticidal, herbicidal and algicidal effects on some organisms (*Artemia salina, Chlorella vulgaris, Daphnia magna, Lactuca sativa*) in aquatic environments (*Harwanto et al., 2022*).

In another study, the insecticidal activity of methanol and chloroform extracts of red algae *Porteria hornemannii* on the fourth stage larvae of three vector mosquitoes; *Culex quinquefasciatus*, *Ae. aegypti* and *An. stephensi* was investigated and larvicidal effects were detected in all extracts applied to all species (Table 2). Furthermore, the methanol extracts were found to have significantly higher toxicity than the chloroform extracts (*Arumugam et al., 2022*). According to the results of GC-MS analysis of *P. hornemannii* methanol extract, it was reported that the substance with 41.88% content was n-hexadecanoic acid, which *Sigamani et al. (2020)* showed as the source of high larvicidal effect on mosquito.

Acetone extract of green algae *Caulerpa scalpelliformis*, which live in tropical to subtropical regions and is considered exotic and invasive on the Turkish coast, has been reported to have toxic effects on late second to early third instar larvae of the common mosquito species *Culex pipiens* (*Cetin, Gokoglu & Oz, 2010*). In a previous study, *Thangam & Kathiresan (1991)* reported that the toxicity of acetone extract of the same species on *Ae. aegypti* larvae was high and the $LC_{50}$ value was 53.7 ppm. In another study, the larvicidal effects of caulerpin and caulerpinic acid isolated from *Caulerpa racemosa* on *Cx. pipiens* mosquito were documented (*Alarif et al., 2010*). Thus, the larvicidal effect of *Caulerpa* sp. species on mosquitoes may be attributed to caulerpin and caulerpinic acid.

*Yu et al. (2015)* reported the mosquitocidal activity of four extracts (chloroform, methanol, n-hexane, and aqueous) of the green algae *Bryopsis pennata* against two vector mosquitoes, *Ae. aegypti* and *Ae. albopictus*. It was pointed out that the larvicidal potential of the chloroform extract was higher than the other extracts (Table 2). The authors also found that the purified chloroform extract of *B. pennata* was fifteen times more effective. When the LC-MS content analysis results given in the article was examined, it was seen that chloroform extracts of algae contain caulerpinic acid and caulerpin derivatives.

Extracts of brown algae *Sargassum wightii* and green algae *Halimeda gracillis* were obtained by *Suganya et al. (2019)* using acetone, chloroform, ethanol, methanol, and water as solvents. These extracts were found to have very high toxicity on the third instar larvae of three different mosquito species (*Anopheles stephensi*, *Ae. aegypti* and *Culex tritaeniorhynchus*) and the authors reported that the $LC_{50}$ values of ethanol extract on the larvae of all species were less than 50 ppm.

It was reported that *Champia parvula* marine algae caused 97% or more mortality on second and third stage larvae of *Ae. aegypti* at a concentration of 100 ppm and $LC_{50}$ and $LC_{90}$ values were 43 ppm and 88 ppm, respectively (*Yogarajalakshmi et al., 2020*). The authors reported that the applied extracts increased the activities of superoxide dismutase, catalase, and glutathione peroxidase enzymes, which are markers of oxidative stress. The larvicidal activity of methanolic extracts of eight algal species (*Jania rubens*, *Galaxaura elongata*, *Gelidium latifolium*—red algae; *Ulva intestinales*, *Codium tomentosum*—green algae; *Dictyota dichotoma*, *Sargassum dentifolium*, *Padina boryana*—brown algae) against third instar larvae of *Cx. pipiens* was investigated by *Haleem et al. (2022)*. The extracts of all algae showed larvicidal activity (Table 2). Similar to *Yogarajalakshmi et al. (2020)*, *Haleem et al. (2022)* were also reported that the activities of oxidative stress markers superoxide dismutase, catalase and glutathione peroxidase enzymes increased. They indicated that the lethal effect may be related to oxidative imbalance resulting from these excessive enzymatic increases.

The insecticidal activity of ethanolic extracts from *Ulva rigida*, *Asparagopsis taxiformis*, *Dictyota dichotoma* and *Cystoseira barbata* algae was tested against *Aedes albopictus* larvae. Only *D. dichotoma* showed activity against *Ae. albopictus* larvae. The $LC_{50}$ and $LC_{90}$ values of ethanolic extracts are 44.32 and 85.92 mg/l, respectively.

In addition, in many toxicity studies with macroalgae (seaweeds), one or more solvents were used simultaneously for extraction. Some of these solvent combinations are petroleum ether-acetone, ethanol-water, methanol-acetone, and dichloromethane-methanol. These studies were conducted on different mosquito species (*Cx. quinquefasciatus*, *Ae. aegypti* and *An. stephensi*) and extracts prepared using solvent combinations were found to be more toxic to mosquitoes (*Thangam & Kathiresan, 1991*; *Manilal et al., 2009*; *Ali, Ravikumar & Beula, 2013*; *Yu et al., 2014*).

## Newer approaches
### Mosquito larvicidal activity of transgenic microalgae
Genetic modification of algae involves the introduction of foreign genes into the algal genome, resulting in transgenic algae. The first step is to identify and isolate the gene of

interest (the foreign gene) that will be introduced into the algae. This gene could be sourced from another organism, such as a plant, animal, or bacteria, or it could be a synthetic gene designed in the laboratory. Once the gene of interest is selected, it needs to be cloned. This involves inserting the gene into a vector, which is a DNA molecule typically derived from a plasmid or a viral genome. The vector acts as a carrier to transport the gene into the algae cells. Transformation methods used for microalgae cells include agitation with glass beads, electroporation, particle bombardment, and *Agrobacterium*-mediated transformation (*Shi et al., 2021*). The selected transgenic algae cells are further verified to ensure that the foreign gene has been integrated into their genome correctly. Molecular techniques such as polymerase chain reaction and DNA sequencing are commonly used for this purpose.

Studies have been conducted on the use of transgenic algal cells in mosquito larvae control, especially with the microalgae genera *Chlamydomonas* and *Chlorella* on *Ae. aegypti* and *Anop. gambia* mosquitoes. In recent studies, the genes of endotoxins (Cry proteins) produced by *Bacillus thuringiensis* subs. *israelensis* (Bti), which are widely used in mosquito larvae control, were transferred to the chloroplast of *Chlamydomonas* to aid in mosquito control (*Kang et al., 2017*). In addition, transgenic *Chlamydomonas reinhardtii* generated by chloroplast expression of the cry11Ba gene were seen to have lethal effects on larvae of *Ae. aegypti* (*Pervaiz et al., 2022*). Since Bti endotoxins are affected by sunlight and degrade quickly in nature, more frequent applications are required in mosquito control. Transferring Bti toxin genes to microalgae will help increase their persistence in the environment. However, as with Cry gene expression in the chloroplast of *Chlamydomonas*, more research is needed to ascertain this special interaction.

Trypsin modulating oostatic factor (TMOF) is an insecticidal protein and an insect peptide hormone synthesized by the mosquito ovary that controls the synthesis of certain enzymes in midgut epithelial cells. After cloning and expressing TMOF in the green alga *Chlorella desiccata,* the preparations were treated with *Ae. aegypti* larvae. It was found that transgenic *C. desiccata* inhibited the translation of trypsin mRNA in intestinal epithelial cells of *Ae. aegypti* larvae. *Ae. aegypti* larvae fed with these transgenic *C. desiccata* cells showed more than 60% mortality within four days (*Borovsky, Sterner & Powell, 2016*).

A comparative study was conducted on a mosquito species fed on transgenic and non-transgenic microalgae. Accordingly, *Anop. gambiae* larvae fed on the transgenic microalgae showed reduced levels of 3-hydroxyquinurenine transaminase (3-HKT) gene expression and higher mortality compared to those fed on non-transgenic algal cells (*Kumar et al., 2013*).

In recent years, ribonucleic acid interference (RNAi) has also been tested in the field of pest control. When *Ae. aegypti* larvae were fed with transgenic *Chlamydomonas* and *Chlorella* microalgae generated by 3-hydroxyquinurenine transaminase (3-HKT) RNAi expression plasmid, the integument and midgut were severely damaged, and the majority of larvae died (*Kang et al., 2017*). *Fei et al. (2020)* and *Fei et al. (2021)* reported that the hormone receptor 3 (HR3) RNAi transgenic microalgae *Chlamydomonas* strains were lethal to *Ae. aegypti* larvae. These authors found that the midgut cavity of treated larvae was disintegrated, and the muscles were unevenly distributed.

Transgenic *Chlorella* microalgae developed by targeting the chitin synthase A (chsa) gene inhibited the development of *Ae. aegypti* under laboratory and semi-field conditions. Another objective of the study was to determine the impact of transgenic algae acting on chitin synthesis on native plankton in aquatic environments. For this purpose, a simulated field release study was conducted, and it was found that the presence of natural phytoplankton and zooplankton species, including wild-type *Chlorella* species, was damaged and disappeared in the aquatic environment where transgenic microalgae acting on chitin synthesis were applied (*Fei et al., 2023a*). In another study, the effect of marker-free RNAi-recombinant transgenic algae (*Chlamydomonas* sp.) on natural plankton was investigated in aquatic environments. As a result, it was reported that the reduction rates of plankton in the environment where transgenic algae were applied did not show a significant difference compared to the area where non-transgenic microalgae were applied (*Fei et al., 2023b*). Although transgenic microalgae have been shown to be highly effective in mosquito control, further studies are needed to investigate their toxic effects on the environment and non-target organisms.

### Nanoparticles of macroalgae (seaweeds) for mosquito larva control

In recent years, there has been an increasing trend toward research on the toxic effects of nanoparticles derived from algae on different insect pests. Some researchers have discovered that macroalgae nanoparticles (such as silver, gold, titanium dioxide, and zinc oxide) can effectively kill mosquito larvae and inhibit their growth (*Deepak et al., 2018*; *Ishwarya et al., 2018*; *Vinoth et al., 2019*). Nanoparticle extracts of macroalgae is prepared by solvent extraction or aqueous extraction. The reaction of the algal extract with the metal salt solution is initiated. The bioactive compounds present in the algal extract act as reducing agents, leading to the reduction of metal ions and the subsequent formation of metal nanoparticles. In the purification and characterization phases, the nanoparticles are separated from the mixture using centrifugation or filtration methods. The purified nanoparticles can be characterized using techniques such as transmission electron microscopy, scanning electron microscopy, atomic force microscopy, dynamic light scattering, X-ray photoelectron spectroscopy, powder X-ray diffractometry, Fourier transform infrared spectroscopy, and UV-vis spectroscopy (*El-Sheekh & El-Kassas, 2016*). Among the macroalgae genera from which nanoparticles have been prepared, *Sargassum* and *Ulva* are the most studied.

*Aziz (2022)* reported that the inhibition of adult emergence in *Ae. aegypti* and *Cx. pipiens* mosquitoes by the crude extract of *Ulva lactuca*, a green macroalgae, and synthesized silver nanoparticles was 97.7% and 93.3%, respectively. In one study, researchers found that seaweed titanium dioxide ($TiO_2$) nanoparticles made from *Sargassum wightii* were highly toxic to *Anop. subpictus* and *Cx. quinquefasciatus* mosquito larvae ($LC_{50}$ values: 4.37 ppm and 4.68 ppm; $LC_{90}$ values: 8.33 ppm and 8.97 ppm), respectively (*Mathivanan et al., 2023*).

Zinc oxide nanoparticles (ZnO NPs) fabricated using *Sarg. wightii* showed higher toxicity than ethanolic extract on four larval instars of *Anop. stephensi*. The $LC_{50}$ values of the ethanolic extract of *S. wightii* on larvae ranged between 57.1–78.7 ppm while the

LC$_{50}$ values of ZnO NPs were 4.3–6.4 ppm. The determined LC$_{50}$ values clearly show that nanoparticles of extracts are much more toxic to mosquito larvae (*Murugan et al., 2018*).

The toxic effect of silver nanoparticles (Ag NPs) prepared using ethanolic extract of *Sarg. palmeri* (SpExt) on the fourth stage larvae of *Ae. aegypti* mosquitoes was investigated, and it was found that extract nanoparticles were 1.45 and 2.77 times more toxic (*Ghramh et al., 2022*).

The *Sarg. muticum* seaweed, which was collected from the Red Sea fabricated silver nanoparticles-induced (Ag NPs) high larval mortality against mosquitoes from both Indian and Saudi Arabian mosquito strains when compared to the seaweed aqueous extract (*Trivedi et al., 2021*).

Larvicidal activity of silver nanoparticles (Ag NPs) of *Sarg. myriocystum* aqueous extract against the fourth instar larvae of mosquitoes *Ae. aegypti* and *Cx. quinquefasciatus* were studied by *Balaraman et al. (2020)*. They reported that silver nanoparticles at 25 ppm caused complete (100%) mortality in both mosquitoes after 48 h of exposure.

## CONCLUSIONS

In conclusion, the literature on the use of micro- and macroalgae in mosquito control has shown promising results. Our review shows that some algae produce compounds that are toxic to mosquitoes and have LC$_{50}$ and LC$_{90}$ values as low as those of many synthetic chemicals. Therefore, we believe that algae can potentially be used as a natural alternative to synthetic insecticides. The toxic effects of algae extracts vary according to the type of extraction, solvent used, combination of solvents, mosquito species, exposure time, larval stage, diversity of algal secondary components, mode of action of components, and interaction between components. Algae can show their toxic effects on mosquitoes through the synergistic effects of the constituents they contain by inhibiting feeding, causing damage to the gut membrane cells, and inhibiting digestion and detoxification enzymes. Toxic effects are not limited to the feeding system. A number of studies have found that some algal extracts also have an effect on the nervous system of larvae, showing signs of unnatural restlessness and wriggling movement within a few hours of application.

In addition, the toxicity of micro and macroalgae components to non-target organisms is much lower than that of many pesticide groups, and their degradation processes in the aquatic environment are very short. However, further research is needed to identify the active compounds and their mechanisms of action, as well as to evaluate their safety and efficacy in field conditions.

### Funding
The authors received no funding for this work.

### Competing Interests
The authors declare there are no competing interests.
## Author Contributions

- Ozge Tufan-Cetin conceived and designed the experiments, performed the experiments, analyzed the data, prepared figures and/or tables, authored or reviewed drafts of the article, prepared manuscript and tables, authored or reviewed drafts of the review, approved the final draft, and approved the final draft.
- Huseyin Cetin conceived and designed the experiments, performed the experiments, analyzed the data, prepared figures and/or tables, authored or reviewed drafts of the article, prepared manuscript and tables, authored or reviewed drafts of the review, approved the final draft, and approved the final draft.

## Data Availability

This is a literature review.

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
