# Peer review of "Use of micro and macroalgae extracts for the control of vector mosquitoes"

_PeerJ, doi:10.7717/peerj.16187_

## Round 0.1 · original submission · Major Revisions

Dear Drs. Tufan-Cetin and Cetin:

Thanks for submitting your review to PeerJ. I have now received two independent reviews of your work, and as you will see, the reviewers raised some concerns about the manuscript. Despite this, these reviewers are optimistic enough about your work and the potential impact it will have on research studying mosquito control measures. Thus, I encourage you to revise your manuscript, accordingly, taking into account all of the concerns raised by both reviewers.

There are many suggestions by the reviewers that should greatly improve your manuscript. Since this is a review, the missing information must be added, or you must address why you choose not to.

Please try to restructure your review for a clearer presentation.

Please edit for clarity, English and grammar. Also, stay consistent between the Introduction and Discussion regarding topics/content. Ensure that your assessments are thoroughly placed within the existing body of literature (topically but also timeframe).

Thus, I encourage you to revise your manuscript, accordingly, taking into account all of the concerns raised by the three reviewers.

Good luck with your revision,

Best,

-joe

Reviewer 1 ·

Basic reporting

Tufan-Cetin & Cetin compiled the literature on the use of micro- and macroalgal extracts, and transgenic microalgae for the control of mosquitoes. This manuscript seems like an interesting approach to the subject matter but its structure and organisation could have been developed better.

The authors claimed to review the "mechanisms" of the extracts but this was only listed as one or two sentences in some paragraphs.

If the extracts are indeed effective and have potential, the next question would be what are (potential) active ingredients? "Compounds with larvicidal activity" appeared numerous times in the manuscript but there seems to be too little information on the identity of these compounds, structure-activity relationship, etc. throughout the manuscript.

Some ideas to improve the content of the manuscript:

1. The subtopic on transgenic microalgae could have been separated from the discussion on the extracts of microalgae.
2. In fact, the aspects of transgenic could have been further developed as it reflects the newer approaches, alongside with the use of nanoparticles, to address this problem.
3. The active ingredients in the extracts that might have contributed to the toxic effects should be discussed.
4. What can be deduced regarding the potency of extracts of microalgae vs. macroalgae vs. transgenic algae?
5. Potential applications of the algal extracts as larvicidal agents. Is it feasible?

Experimental design

The survey methodology is acceptable.
Please state the range for year of publications of the articles used in this study.
Did the authors consider publications that are not in English? Otherwise, this review is not comprehensive and reflect the actual developments in this field.
The review should be organised in more coherent paragraphs and subsections (refer above).

Validity of the findings

Extensive information has been compiled and tabulated, however, critical analysis and synthesis of information from those information is still lacking. For instance, the authors merely listed the toxic effects of extracts of different algae species but no conclusions were drawn from those data. Similarly, several factors may affect the toxicity of the extracts towards larvae were listed but the authors did not conclude how each factor, based on their summary, influence the toxicity - it is apparent that different extraction solvents capture different array of compounds without even looking at the related literature, one would, therefore, be interested to know what types/classes of compounds are effective?

Toxicity of the extracts were tabulated but the different authors might have carried the experiments under different conditions and the LC50 might have been calculated using different methods. Did the authors take this into account?

Future perspectives based on the authors' experience and opinion are crucial component of a good review manuscript.

Additional comments

The subject matter is interesting but its content must be improved to incorporate critical analysis and synthesis of the relevant literature. This manuscript also lacks a number of subtopics (some suggestions are listed above) which could otherwise elevate the impact of this work.

·

Basic reporting

- The paper needs grammatical and English revision.
- The availability, distribution and application of micro and macroalgae should be mentioned in the introduction.

Experimental design

- The types of solvents and methods of algal extraction should be investigated.
- The method of genetic modification to produce transgenic algae should be mentioned.
- The main concept of nano algal extract preparation should be explained.

Validity of the findings

- The introduction focused on the larvicidal activity of micro and macroalgae while the results and discussion section was mainly based on transgenic algae.
- The comparison between the larvicidal activity of algal extracts and transgenic algae should be clear.
- The mode of action of algal extracts and their various impacts on larvae should be explained clearly.

---

## Round 0.2 · Minor Revisions

Dear Drs. Tufan-Cetin and Cetin:

Thanks for revising your manuscript. The reviewers are mostly satisfied with your revision (as am I). Great! However, there are a few additional concerns to address per reviewer 1. Please attend to these issues ASAP so we may move towards acceptance of your work.

Best,

-joe

Reviewer 1 ·

Basic reporting

The authors have responded to most of the reviewers' comments and edited the manuscript accordingly. The authors tried to improve the manuscript but I am of opinion that the bulk of the manuscript is rather descriptive, for instance, the larvicidal activity of the extracts were compiled in two tables (Table 1 & 2) yet in the text there are lengthy descriptions of each of the study that made the section rather repetitive and redundant (in view of existing tables on those info). There are only about 10 previously published work on that area (as shown in Tables 1 & 2) hence, this leaves an impression that this work is rather preliminary. A review on similar scope was published almost ten years ago (Yu et al. 2014) but it is unclear how this present manuscript may bridge our understanding in this topic, for instance, there is too little information on the chemistry aspect. Also, graphical representation of the mechanisms, newer approaches etc. may be useful. These and other improvements are therefore recommended to elevate the impact of this manuscript - if it is accepted.

Experimental design

No comment

Validity of the findings

No comment

·

Basic reporting

The topic has become more comprehensive and covers different aspects

Experimental design

The study Design highly enhanced and be clear

Validity of the findings

No more comments

---

## Round 0.3 · accepted · Accept

Dear Drs. Tufan-Cetin and Cetin:

Thanks for revising your manuscript based on the concerns raised by the reviewers. I now believe that your manuscript is suitable for publication. Congratulations! I look forward to seeing this work in print, and I anticipate it being an important resource for groups studying mosquito control measures. Thanks again for choosing PeerJ to publish such important work.

Best,

-joe